# SMOOTHING THE GEOMETRY
# OF PROBABILISTIC BOX EMBEDDINGS

**Xiang Li**[*]**, Luke Vilnis**[*]**, Dongxu Zhang, Michael Boratko & Andrew McCallum**
College of Information and Computer Sciences
University of Massachusetts Amherst

## ABSTRACT

There is growing interest in geometrically-inspired embeddings for learning hier-
archies, partial orders, and lattice structures, with natural applications to transitive
relational data such as entailment graphs. Recent work has extended these ideas
beyond deterministic hierarchies to probabilistically calibrated models, which en-
able learning from uncertain supervision and inferring soft-inclusions among con-
cepts, while maintaining the geometric inductive bias of hierarchical embedding
models. We build on the Box Lattice model of Vilnis et al. (2018), which showed
promising results in modeling soft-inclusions through an overlapping hierarchy of
sets, parameterized as high-dimensional hyperrectangles (boxes). However, the
hard edges of the boxes present difficulties for standard gradient based optimiza-
tion; that work employed a special surrogate function for the disjoint case, but
we find this method to be fragile. In this work, we present a novel hierarchical
embedding model, inspired by a relaxation of box embeddings into parameter-
ized density functions using Gaussian convolutions over the boxes. Our approach
provides an alternative surrogate to the original lattice measure that improves the
robustness of optimization in the disjoint case, while also preserving the desir-
able properties with respect to the original lattice. We demonstrate increased or
matching performance on WordNet hypernymy prediction, Flickr caption entail-
ment and a MovieLens-based market basket dataset. We show especially marked
improvements in the case of sparse data, where many conditional probabilities
should be low, and thus boxes should be nearly disjoint.

## 1 INTRODUCTION

Embedding methods have long been a key technique in machine learning, providing a natural way
to convert semantic problems into geometric problems. Early examples include the vector space
(Salton et al., 1975) and latent semantic indexing (Deerwester et al., 1990) models for information
retrieval. Embeddings experienced a renaissance after the publication of Word2Vec (Mikolov et al.,
2013), a neural word embedding method (Bengio et al., 2003; Mnih & Hinton, 2009) that could run
at massive scale.

Recent years have seen an interest in *structured* or *geometric* representations. Instead of represent-
ing e.g. images, words, sentences, or knowledge base concepts with points, these methods instead
associate them with more complex geometric structures. These objects can be density functions, as
in Gaussian embeddings (Vilnis & McCallum, 2015; Athiwaratkun & Wilson, 2017; 2018), convex
cones, as in order embeddings (Vendrov et al., 2016; Lai & Hockenmaier, 2017), or axis-aligned hy-
perrectangles, as in box embeddings (Vilnis et al., 2018; Subramanian & Chakrabarti, 2018). These
geometric objects more naturally express ideas of asymmetry, entailment, ordering, and transitive
relations than simple points in a vector space, and provide a strong inductive bias for these tasks.

In this work, we focus on the probabilistic Box Lattice model of Vilnis et al. (2018), because of its
strong empirical performance in modeling transitive relations, probabilistic interpretation (edges in
a relational DAG are replaced with conditional probabilities), and ability to model complex joint

---

[*] Equal contribution.

probability distributions including negative correlations. Box embeddings (BE) are a generalization of order embeddings (OE) (Vendrov et al., 2016) and probabilistic order embeddings (POE) (Lai & Hockenmaier, 2017) that replace the vector lattice ordering (notions of overlapping and enclosing convex cones) in OE and POE with a more general notion of overlapping boxes (products of intervals).

While intuitively appealing, the "hard edges" of boxes and their ability to become easily disjoint, present difficulties for gradient-based optimization: when two boxes are disjoint in the model, but have overlap in the ground truth, no gradient can flow to the model to correct the problem. This is of special concern for (pseudo-)sparse data, where many boxes should have nearly zero overlap, while others should have very high overlap. This is especially pronounced in the case of e.g. market basket models for recommendation, where most items should not be recommended, and entailment tasks, most of which are currently artificially resampled into a 1:1 ratio of positive to negative examples. To address the disjoint case, Vilnis et al. (2018) introduce an ad-hoc surrogate function. In contrast, we look at this problem as inspiration for a new model, based on the intuition of relaxing the hard edges of the boxes into smoothed density functions, using a Gaussian convolution with the original boxes.

We demonstrate the superiority of our approach to modeling transitive relations on WordNet, Flickr caption entailment, and a MovieLens-based market basket dataset. We match or beat existing state of the art results, while showing substantial improvements in the pseudosparse regime.

## 2 Related Work

As mentioned in the introduction, there is much related work on structured or geometric embeddings. Most relevant to this work are the order embeddings of Vendrov et al. (2016), which embed a non-probabilistic DAG or lattice in a vector space with order given by inclusion of embeddings' forward cones, the probabilistic extension of that model due to Lai & Hockenmaier (2017), and the *box lattice* or *box embedding* model of Vilnis et al. (2018), which we extend. Concurrently to Vilnis et al. (2018), another hyperrectangle-based generalization of order embeddings was proposed by Subramanian & Chakrabarti (2018), also called *box embeddings*. The difference between the two models lies in the interpretation: the former is a probabilistic model that assigns edges conditional probabilities according to degrees of overlap, while the latter is a deterministic model in the style of order embeddings — an edge is considered present only if one box entirely encloses another.

Methods based on embedding points in hyperbolic space (Nickel & Kiela, 2017; Ganea et al., 2018) have also recently been proposed for learning hierarchical embeddings. These models, similar to order embeddings and the box embeddings of Subramanian & Chakrabarti (2018), are non-probabilistic and optimize an energy function. Additionally, while the negative curvature of hyperbolic space is attractively biased towards learning tree structures (since distances between points increase the farther they are from the origin), this constant curvature makes the models not as suitable for learning non-treelike DAGs.

Our approach to smoothing the energy landscape of the model using Gaussian convolution is common in mollified optimization and continuation methods, and is increasingly making its way into machine learning models such as Mollifying Networks (Gulcehre et al., 2016b), diffusion-trained networks (Mobahi, 2016), and noisy activation functions (Gulcehre et al., 2016a).

Our focus on embedding orderings and transitive relations is a subset of knowledge graph embedding. While this field is very large, the main difference of our probabilistic approach is that we seek to learn an embedding model which maps concepts to subsets of event space, giving our model an inductive bias especially suited for transitive relations as well as fuzzy concepts of inclusion and entailment.

## 3 Background

We begin with a brief overview of two methods for representing ontologies as geometric objects. First, we review some definitions from order theory, a useful formalism for describing ontologies, then we introduce the vector and box lattices. Figure 1 shows a simple two-dimensional example of these representations.

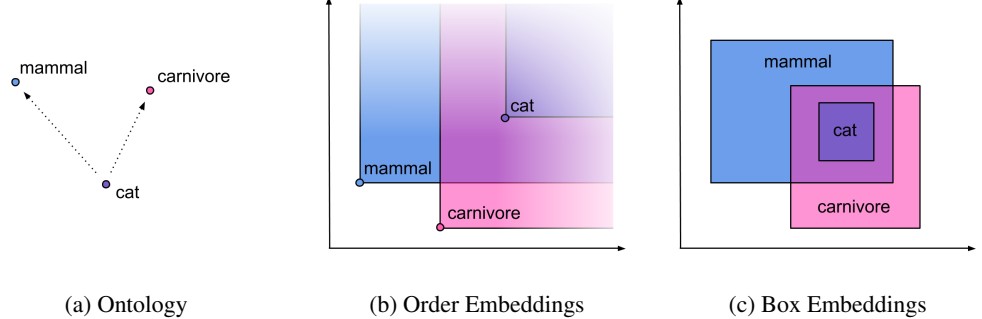

| (a) Ontology | (b) Order Embeddings | (c) Box Embeddings |

Figure 1: Comparison between the Order Embedding (vector lattice) and Box Embedding representations for a simple ontology. Regions represent concepts and overlaps represent their entailment. Shading represents density in the probabilistic case.

### 3.1 PARTIAL ORDERS AND LATTICES

A non-strict *partially ordered set* (*poset*) is a pair $P, \preceq$, where $P$ is a set, and $\preceq$ is a binary relation. For all $a, b, c \in P$,

**Reflexivity**: $a \preceq a$

**Antisymmetry**: $a \preceq b \preceq a$ implies $a = b$

**Transitivity**: $a \preceq b \preceq c$ implies $a \preceq c$

This generalizes the standard concept of a totally ordered set to allow some elements to be incomparable. Posets provide a good formalism for the kind of acyclic directed graph data found in many knowledge bases with transitive relations.

A *lattice* is a poset where any subset of elements has a single unique least upper bound, and greatest lower bound. In a *bounded lattice*, the set $P$ contains two additional elements, $\top$ (*top*), and $\perp$ (*bottom*), which denote the least upper bound and greatest lower bound of the entire set.

A lattice is equipped with two binary operations, $\vee$ (*join*), and $\wedge$ (*meet*). $a \vee b$ denotes the least upper bound of $a, b \in P$, and $a \wedge b$ denotes their greatest lower bound. A bounded lattice must satisfy these properties:

**Idempotency**: $a \wedge a = a \vee a = a$

**Commutativity**: $a \wedge b = b \wedge a$ and $a \vee b = b \vee a$

**Associativity**: $a \wedge b \wedge c = a \wedge (b \wedge c)$ and $(a \vee b \vee c) = a \vee (b \vee c)$

**Absorption**: $a \vee (a \wedge b) = a$ and $a \wedge (a \vee b) = a$

**Bounded**: $\perp \preceq a \preceq \top$

Note that the extended real numbers, $\mathbb{R} \cup \{-\infty, \infty\}$, form a bounded lattice (and in fact, a totally ordered set) under the $\min$ and $\max$ operations as the meet ($\wedge$) and join ($\vee$) operations. So do sets partially ordered by inclusion, with $\cap$ and $\cup$ as $\wedge$ and $\vee$. Thinking of these special cases gives the intuition for the fourth property, *absorption*.

The $\wedge$ and $\vee$ operations can be swapped, along with reversing the poset relation $\preceq$, to give a valid lattice, called the *dual lattice*. In the real numbers this just corresponds to a sign change. A *semilattice* has only a meet or join, but not both.

**Note.** In the rest of the paper, when the context is clear, we will also use $\wedge$ and $\vee$ to denote $\min$ and $\max$ of real numbers, in order to clarify the intuition behind our model.

### 3.2 VECTOR LATTICE

A *vector lattice*, also known as a *Riesz space* (Zaanen, 1997), or *Hilbert lattice* when the accompanying vector space has an inner product, is a vector space endowed with a lattice structure.

A standard choice of partial order for the vector lattice $\mathbb{R}^n$ is to use the *product order* from the underlying real numbers, which specifies for all $\mathbf{x}, \mathbf{y} \in \mathbb{R}^n$

$$\mathbf{x} \preceq \mathbf{y} \iff \forall i \in \{1..n\}, \ x_i \leq y_i$$

Under this order, meet and join operations are pointwise $\min$ and $\max$, which gives a lattice structure. In this formalism, the Order Embeddings of Vendrov et al. (2016) embed partial orders as vectors using the *reverse* product order, corresponding to the dual lattice, and restrict the vectors to be positive. The vector of all zeroes represents $\top$, and embedded objects become "more specific" as they get farther away from the origin.

Figure 1b demonstrates a toy, two-dimensional example of the Order Embedding vector lattice representation of a simple ontology. Shading represents the probability measure assigned to this lattice in the probabilistic extension of Lai & Hockenmaier (2017).

## 3.3 Box Lattice

Vilnis et al. (2018) introduced a *box lattice*, wherein each concept in a knowledge graph is associated with two vectors, the minimum and maximum coordinates of an axis-aligned hyperrectangle, or *box* (product of intervals).

Using the notion of set inclusion between boxes, there is a natural partial order and lattice structure. To represent a box $\mathbf{x}$, let the pairs $(x_{m,i}, x_{M,i})$ be the maximum and minimum of the interval at each coordinate $i$. Then the box lattice structure (least upper bounds and greatest lower bounds), with $\vee$ and $\wedge$ denoting $\max$ and $\min$ when applied to the scalar coordinates, is

$$\mathbf{x} \wedge \mathbf{y} = \bot \ \text{ if } x \text{ and } y \text{ disjoint, else}$$

$$\mathbf{x} \wedge \mathbf{y} = \prod_i [x_{m,i} \vee y_{m,i}, x_{M,i} \wedge y_{M,i}]$$

$$\mathbf{x} \vee \mathbf{y} = \prod_i [x_{m,i} \wedge y_{m,i}, x_{M,i} \vee y_{M,i}]$$

Here, $\prod$ denotes a set (cartesian) product — the lattice meet is the largest box contained entirely within both $\mathbf{x}$ and $\mathbf{y}$, or bottom (the empty set) where no intersection exists, and the lattice join is the smallest box containing both $\mathbf{x}$ and $\mathbf{y}$.

To associate a measure, marginal probabilities of (collections of) events are given by the volume of boxes, their complements, and intersections under a suitable probability measure. Under the uniform measure, if event $\mathbf{x}$ has an associated box with interval boundaries $(x_m, x_M)$, the probability $p(\mathbf{x})$ is given by $\prod_i^n (x_{M,i} - x_{m,i})$. Use of the uniform measure requires the boxes to be constrained to the unit hypercube, so that $p(\mathbf{x}) \leq 1$. $p(\bot)$ is taken to be zero, since $\bot$ is an empty set. As boxes are simply special cases of sets, it is intuitive that this is a valid probability measure, but it can also be shown to be compatible with the meet semilattice structure in a precise sense (Leader, 1971).

Figure 1c demonstrates a toy, two-dimensional example of the Box Embedding lattice representation of a simple ontology.

## 4 Method

### 4.1 Motivation: Optimization and Sparse Data

When using gradient-based optimization to learn box embeddings, an immediate problem identified in the original work is that when two concepts are incorrectly given as disjoint by the model, no gradient signal can flow since the meet (intersection) is exactly zero, with zero derivative. To see this, note that for a pair of 1-dimensional boxes (intervals), the volume of the meet under the uniform measure $p$ as given in Section 3.3 is

$$p(\mathbf{x} \wedge \mathbf{y}) = m_h(\min(x_M, y_M) - \max(x_m, y_m)) \tag{1}$$

where $m_h$ is the standard hinge function, $m_h(x) = 0 \vee x = \max(0, x)$.

The hinge function has a large flat plateau at $0$ when intervals are disjoint. This issue is especially problematic when the lattice to be embedded is (pseudo-)sparse, that is, most boxes should have very little or no intersection, since if training accidentally makes two boxes disjoint there is no way to recover with the naive measure. The authors propose a surrogate function to optimize in this case, but we will use a more principled framework to develop alternate measures that avoid this pathology, improving both optimization and final model quality.

## 4.2 RELAXED GEOMETRY

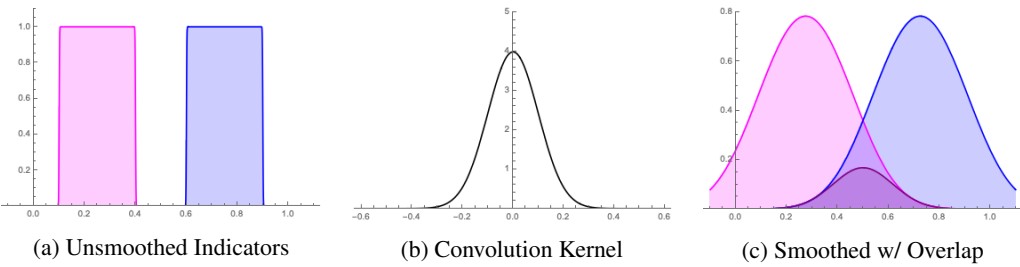

(a) Unsmoothed Indicators      (b) Convolution Kernel      (c) Smoothed w/ Overlap

Figure 2: One-dimensional example demonstrating two disjoint indicators of intervals before and after the application of a smoothing kernel. The area under the purple product curve is proportional to the degree of overlap.

The intuition behind our approach is that the "hard edges" of the standard box embeddings lead to unwanted gradient sparsity, and we seek a relaxation of this assumption that maintains the desirable properties of the base lattice model while enabling better optimization and preserving a geometric intuition. For ease of exposition, we will refer to 1-dimensional intervals in this section, but the results carry through from the representation of boxes as products of intervals and their volumes under the associated product measures.

The first observation is that, considering boxes as indicator functions of intervals, we can rewrite the measure of the joint probability $p(\mathbf{x} \wedge \mathbf{y})$ between intervals $\mathbf{x} = [a, b]$ and $\mathbf{y} = [c, d]$ as an integral of the product of those indicators:

$$p(\mathbf{x} \wedge \mathbf{y}) = \int_{\mathbb{R}} \mathbb{1}_{[a,b]}(x)\mathbb{1}_{[c,d]}(x)dx$$

since the product has support (and is equal to 1) only in the areas where the two intervals overlap.

A solution suggests itself in replacing these indicator functions with functions of infinite support. We elect for kernel smoothing, specifically convolution with a normalized Gaussian kernel, equivalent to an application of the diffusion equation to the original functional form of the embeddings (indicator functions) and a common approach to mollified optimization and energy smoothing (Neelakantan et al., 2015; Gulcehre et al., 2016b; Mobahi, 2016). This approach is demonstrated in one dimension in Figure 2.

Specifically, given $\mathbf{x} = [a, b]$, we associate the smoothed indicator function

$$f(x; a, b, \sigma^2) = \mathbb{1}_{[a,b]}(x) * \phi(x; \sigma^2) = \int_{\mathbb{R}} \mathbb{1}_{[a,b]}(z)\phi(x - z; \sigma^2)dz = \int_{a}^{b} \phi(x - z; \sigma^2)dz$$

We then wish to evaluate, for two lattice elements $\mathbf{x}$ and $\mathbf{y}$ with associated smoothed indicators $f$ and $g$,

$$p_\phi(\mathbf{x} \wedge \mathbf{y}) = \int_{\mathbb{R}} f(x; a, b, \sigma_1^2)g(x; c, d, \sigma_2^2)dx \tag{2}$$

This integral admits a closed form solution.

**Proposition 1.** *Let $m_\Phi(x) = \int \Phi(x)dx$ be an antiderivative of the standard normal CDF. Then the solution to equation 2 is given by,*

$$p_\phi(\mathbf{x} \wedge \mathbf{y}) = \sigma \left( m_\Phi(\tfrac{b-c}{\sigma}) + m_\Phi(\tfrac{a-d}{\sigma}) - m_\Phi(\tfrac{b-d}{\sigma}) - m_\Phi(\tfrac{a-c}{\sigma}) \right) \tag{3}$$

$$\approx \left( \rho\,\mathrm{soft}(\tfrac{b-c}{\rho}) + \rho\,\mathrm{soft}(\tfrac{a-d}{\rho}) \right) - \left( \rho\,\mathrm{soft}(\tfrac{b-d}{\rho}) + \rho\,\mathrm{soft}(\tfrac{a-c}{\rho}) \right) \tag{4}$$

*where $\sigma = \sqrt{\sigma_1^2 + \sigma_2^2}$, $\mathrm{soft}(x) = \log(1 + \exp(x))$ is the softplus function, the antiderivative of the logistic sigmoid, and $\rho = \frac{\sigma}{1.702}$.*

*Proof.* The first line is proved in Appendix A, the second approximation follows from the approximation of $\Phi$ by a logistic sigmoid given in Bowling et al. (2009). □

Note that, in the *zero-temperature limit*, as $\rho$ goes to zero, we recover the formula

$$p_\phi(\mathbf{x} \wedge \mathbf{y}) = \lim_{\rho \to 0} \left( \rho\,\mathrm{soft}(\tfrac{b-c}{\rho}) + \rho\,\mathrm{soft}(\tfrac{a-d}{\rho}) \right) - \left( \rho\,\mathrm{soft}(\tfrac{b-d}{\rho}) + \rho\,\mathrm{soft}(\tfrac{a-c}{\rho}) \right)$$
$$= \left( m_h(b - c) + m_h(a - d) \right) - \left( m_h(b - d) + m_h(a - c) \right)$$
$$= m_h(b \wedge d - a \vee c)$$

with equality in the last line because $(a, b)$ and $(c, d)$ are intervals. This last line is exactly our original equation equation 1, which is expected from convolution with a zero-bandwidth kernel (a Dirac delta function, the identity element under convolution). This is true for both the exact formula using $\int \Phi(x)dx$, and the softplus approximation.

Unfortunately, for any $\rho > 0$, multiplication of Gaussian-smoothed indicators does not give a valid meet operation on a function lattice, for the simple reason that $f^2 \neq f$, except in the case of indicator functions, violating the idempotency requirement of Section 3.1.

More importantly, for practical considerations, if we are to treat the outputs of $p_\phi$ as probabilities, the consequence is

$$p_\phi(\mathbf{x}|\mathbf{x}) = \frac{p_\phi(\mathbf{x}, \mathbf{x})}{p_\phi(\mathbf{x})} = \frac{p_\phi(\mathbf{x} \wedge \mathbf{x})}{p_\phi(\mathbf{x})} \neq 1 \tag{5}$$

which complicates our applications that train on conditional probabilities. However, by a modification of equation 3, we can obtain a function $p$ such that $p(\mathbf{x} \wedge \mathbf{x}) = p(\mathbf{x})$, while retaining the smooth optimization properties of the Gaussian model.

Recall that for the hinge function $m_h$ and two intervals $(a, b)$ and $(c, d)$, we have

$$\left( m_h(b - c) + m_h(a - d) \right) - \left( m_h(b - d) + m_h(a - c) \right) = m_h(b \wedge d - a \vee c) \tag{6}$$

where the left hand side is the zero-temperature limit of the Gaussian model from equation 3. This identity is true of the hinge function $m_h$, but not the softplus function.

However, an equation with a similar functional form as equation 6 (on both the left- and right-hand sides) is true not only of the hinge function from the unsmoothed model, but also true of the softplus. For two intervals $\mathbf{x} = (a, b)$ an $\mathbf{y} = (c, d)$, by the commutativity of $\min$ and $\max$ with monotonic functions, we have

$$\left( \mathrm{soft}(b - c) \vee \mathrm{soft}(a - d) \right) \wedge \left( \mathrm{soft}(b - d) \vee \mathrm{soft}(a - c) \right) = \mathrm{soft}(b \wedge d - a \vee c) \tag{7}$$

In the zero-temperature limit, all terms in equations 3 and 7 are equivalent. However, outside of this, equation 7 is idempotent for $\mathbf{x} = \mathbf{y} = (a, b) = (c, d)$ (when considered as a measure of overlap, made precise in the next paragraph), while equation 3 is not.

This inspires us to define the probabilities $p(\mathbf{x})$ and $p(\mathbf{x}, \mathbf{y})$ using a normalized version of equation 7 in place of equation 3. For the interval (one-dimensional box) case, we define

$$p(\mathbf{x}) \propto \mathrm{soft}(b - a)$$
$$p(\mathbf{x}, \mathbf{y}) \propto \mathrm{soft}(b \wedge d - a \vee c)$$

which satisfies the idempotency requirement, $p(\mathbf{x}) = p(\mathbf{x}, \mathbf{x})$.

Because softplus upper-bounds the hinge function, it is capable of outputting values that are greater than 1, and therefore must be normalized. In our experiments, we use two different approaches to

normalization. For experiments with a relatively small number of entities (all besides Flickr), we allow the boxes to learn unconstrained, and divide each dimension by the measured size of the global minimum and maximum $(G_m^{(i)}, G_M^{(i)})$ at that dimension

$$m_{\text{soft}}^{(i)}(x) = \frac{\text{soft}(\frac{x}{\rho})}{\text{soft}(\frac{G_m - G_m}{\rho})}$$

For data where computing these values repeatedly is infeasible, we project onto the unit hypercube and normalize by $m_{\text{soft}}(1)$. The final probability $p(\mathbf{x})$ is given by the product over dimensions

$$p(\mathbf{x}) = \prod_i m_{\text{soft}}^{(i)}(x_{M,i} - x_{m,i})$$

$$p(\mathbf{x}, \mathbf{y}) = \prod_i m_{\text{soft}}^{(i)}(x_{M,i} \wedge y_{M,i} - x_{m,i} \vee y_{m,i})$$

Note that, while equivalent in the zero temperature limit to the standard uniform probability measure of the box model, this function, like the Gaussian model, is not a valid probability measure on the entire joint space of events (the lattice). However, neither is factorization of a conditional probability table using a logistic sigmoid link function, which is commonly used for the similar tasks. Our approach retains the inductive bias of the original box model, is equivalent in the limit, and satisfies the necessary condition that $p(\mathbf{x}, \mathbf{x}) = p(\mathbf{x})$. A comparison of the 3 different functions is given in Figure 3, with the softplus overlap showing much better behavior for highly disjoint boxes than the Gaussian model, while also preserving the meet property.

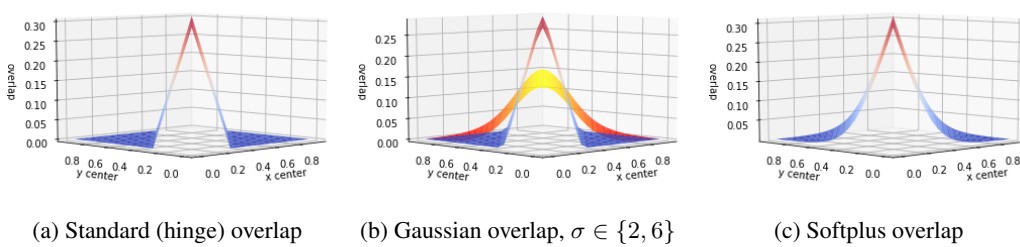

(a) Standard (hinge) overlap   (b) Gaussian overlap, $\sigma \in \{2, 6\}$   (c) Softplus overlap

Figure 3: Comparison of different overlap functions for two boxes of width 0.3 as a function of their centers. Note that in order to achieve high overlap, the Gaussian model must drastically lower its temperature, causing vanishing gradients in the tails.

## 5 EXPERIMENTS

### 5.1 WORDNET

| Method | Test Accuracy % |
|---|---|
| transitive | 88.2 |
| word2gauss | 86.6 |
| OE | 90.6 |
| Li et al. (2017) | 91.3 |
| POE | 91.6 |
| Box | 92.2 |
| Smoothed Box | 92.0 |

Table 4: Classification accuracy on WordNet test set.

We perform experiments on the WordNet hypernym prediction task in order to evaluate the performance of these improvements in practice. The WordNet hypernym hierarchy contains 837,888-edges after performing the transitive closure on the direct edges in WordNet. We used the same train/dev/test split as in Vendrov et al. (2016). Positive examples are randomly chosen from the

837k edges, while negative examples are generated by swapping one of the terms to a random word in the dictionary. Experimental details are given in Appendix D.1.

The smoothed box model performs nearly as well as the original box lattice in terms of test accuracy[1]. While our model requires less hyper-parameter tuning than the original, we suspect that our performance would be increased on a task with a higher degree of sparsity than the 50/50 positive/negative split of the standard WordNet data, which we explore in the next section.

## 5.2 IMBALANCED WORDNET

In order to confirm our intuition that the smoothed box model performs better in the sparse regime, we perform further experiments using different numbers of positive and negative examples from the WordNet *mammal* subset, comparing the box lattice, our smoothed approach, and order embeddings (OE) as a baseline. The training data is the transitive reduction of this subset of the *mammal* Word-Net, while the dev/test is the transitive closure of the training data. The training data contains 1,176 positive examples, and the dev and test sets contain 209 positive examples. Negative examples are generated randomly using the ratio stated in the table.

As we can see from the table, with balanced data, all models include OE baseline, Box, Smoothed Box models nearly match the full transitive closure. As the number of negative examples increases, the performance drops for the original box model, but Smoothed Box still outperforms OE and Box in all setting. This superior performance on imbalanced data is important for e.g. real-world entailment graph learning, where the number of negatives greatly outweigh the positives.

| Positive:Negative | Box | OE | Smoothed Box |
|---|---|---|---|
| 1:1 | 0.9905 | 0.9976 | **1.0** |
| 1:2 | 0.8982 | 0.9139 | **1.0** |
| 1:6 | 0.6680 | 0.6640 | **0.9561** |
| 1:10 | 0.5495 | 0.5897 | **0.8800** |

Table 5: F1 scores of the box lattice, order embeddings, and our smoothed model, for different levels of label imbalance on the WordNet *mammal* subset.

## 5.3 FLICKR

We conduct experiments on the Flickr entailment dataset. Flickr is a large-scale caption entailment dataset containing of 45 million image caption pairs. In order to perform an apples-to-apples comparison with existing results we use the exact same dataset from Vilnis et al. (2018). In this case, we do constrain the boxes to the unit cube, using the same experimental setup as Vilnis et al. (2018), except we apply the softplus function before calculating the volume of the boxes. Experimental details are given in Appendix D.3.

We report KL divergence and Pearson correlation on the full test data, unseen pairs (caption pairs which are never occur in training data) and unseen captions (captions which are never occur in training data). As shown in Table 6, we see a slight performance gain compared to the original model, with improvements most concentrated on unseen captions.

## 5.4 MOVIELENS

We apply our method to a market-basket task constructed using the MovieLens dataset. Here, the task is to predict users' preference for movie A given that they liked movie B. We first collect all pairs of user-movie ratings higher than 4 points (strong preference) from the MovieLens-20M dataset. From this we further prune to just a subset of movies which have more than 100 user ratings to make sure that counting statistics are significant enough. This leads to 8545 movies in our dataset. We calculate the conditional probability $P(A|B) = \frac{P(A,B)}{P(B)} = \frac{\#rating(A,B)_{>4}/\#users}{\#rating(B)_{>4}/\#users}$. We

---

[1] Accuracy is calculated by applying the same threshold which maximized accuracy in dev set.

| Full test data | $P(x\|y)$ KL | Pearson R |
|---|---|---|
| POE | 0.031 | 0.949 |
| POE* | 0.031 | 0.949 |
| Box | 0.020 | 0.967 |
| Smoothed Box | **0.018** | **0.969** |
| *Unseen pairs* | | |
| POE | 0.048 | 0.920 |
| POE* | 0.046 | 0.925 |
| Box | 0.025 | **0.957** |
| Smoothed Box | **0.024** | **0.957** |
| *Unseen captions* | | |
| POE | 0.127 | 0.696 |
| POE* | 0.084 | 0.854 |
| Box | 0.050 | 0.900 |
| Smoothed Box | **0.036** | **0.917** |

Table 6: KL and Pearson correlation between model and gold probability.

randomly pick 100K conditional probabilities for training data and 10k probabilities for dev and test data [2].

We compare with several baselines: low-rank matrix factorization, complex bilinear factorization (Trouillon et al., 2016), and two hierarchical embedding methods, POE (Lai & Hockenmaier, 2017) and the Box Lattice (Vilnis et al., 2018). Since the training matrix is asymmetric, we used separate embeddings for target and conditioned movies. For the complex bilinear model, we added one additional vector of parameters to capture the "imply" relation. We evaluate on the test set using KL divergence, Pearson correlation, and Spearman correlation with the ground truth probabilities. Experimental details are given in Appendix D.4.

From the results in Table 7, we can see that our smoothed box embedding method outperforms the original box lattice as well as all other baselines' performances, especially in Spearman correlation, the most relevant metric for recommendation, a ranking task. We perform an additional study on the robustness of the smoothed model to initialization conditions in Appendix C.

| | KL | Pearson R | Spearman R |
|---|---|---|---|
| Matrix Factorization | 0.0173 | 0.8549 | 0.8374 |
| Complex Bilinear Factorization | 0.0141 | 0.8771 | 0.8636 |
| POE | 0.0170 | 0.8548 | 0.8511 |
| Box | 0.0147 | 0.8775 | 0.8768 |
| Smoothed Box | **0.0138** | **0.8985** | **0.8977** |

Table 7: Performance of the smoothed model, the original box model, and several baselines on MovieLens.

## 6 CONCLUSION AND FUTURE WORK

We presented an approach to smoothing the energy and optimization landscape of probabilistic box embeddings and provided a theoretical justification for the smoothing. Due to a decreased number of hyper-parameters this model is easier to train, and, furthermore, met or surpassed current state-of-the-art results on several interesting datasets. We further demonstrated that this model is particularly effective in the case of sparse data and more robust to poor initialization.

Tackling the learning problems presented by rich, geometrically-inspired embedding models is an open and challenging area of research, which this work is far from the last word on. This task will become even more pressing as the embedding structures become more complex, such as unions of

---

[2]In the dev and test data, we also remove all the $P(A|B)$ where $P(B|A)$ appears in the training data.

boxes or other non-convex objects. To this end, we will continue to explore both function lattices, and constraint-based approaches to learning.

## 7 ACKNOWLEDGMENTS

We thank Travis Wolfe, Colin Evans, Rob Zinkov, Ben Poole, and Laurent Dinh for helpful discussions. We also thank the anonymous reviewers for their constructive feedback. This work was supported in part by the Center for Intelligent Information Retrieval and the Center for Data Science, in part by the Chan Zuckerberg Initiative under the project Scientific Knowledge Base Construction, and in part by the National Science Foundation under Grant No. IIS-1514053. Any opinions, findings and conclusions or recommendations expressed in this material are those of the authors and do not necessarily reflect those of the sponsor.

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

# Supplementary Material

## A    PROOF OF GAUSSIAN OVERLAP FORMULA

We wish to evaluate, for two lattice elements $\mathbf{x}$ and $\mathbf{y}$, with associated smoothed indicators $f$ and $g$,

$$f(x; a, b, \sigma^2) = \mathbb{1}_{[a,b]}(x) * \phi(x; \sigma^2) = \int_{\mathbb{R}} \mathbb{1}_{[a,b]}(z)\phi(x - z; \sigma^2)dz = \int_a^b \phi(x - z; \sigma^2)dz$$

$$p_\phi(\mathbf{x} \wedge \mathbf{y}) = \int_{\mathbb{R}} f(x; a, b, \sigma_1^2)g(x; c, d, \sigma_2^2)dx \tag{8}$$

Since the Gaussian kernel is normalized to have total integral equal to 1, so as not to change the overall areas of the boxes, the concrete formula is

$$\phi(z; \sigma^2) = \frac{1}{\sigma\sqrt{2\pi}}e^{\frac{-z^2}{2\sigma^2}}$$

Since the antiderivative of $\phi$ is the normal CDF, this may be recognized as the difference $\Phi(x; a, \sigma^2) - \Phi(x; b, \sigma^2)$, but this does not allow us to easily evaluate the integral of interest, which is the integral of the product of two such functions.

To evaluate equation 8, recall the identity (Jebara et al., 2004; Vilnis & McCallum, 2015)

$$\int_{\mathbb{R}} \phi(x - \mu_1; \sigma_1^2)\phi(x - \mu_2; \sigma_2^2)dx = \phi(\mu_1 - \mu_2; \sigma_1^2 + \sigma_2^2) \tag{9}$$

For convenience, let $\tau := \frac{1}{\sqrt{\sigma_1^2 + \sigma_2^2}}$. Applying Fubini's theorem and using equation 9, we have

$$\begin{aligned}
p_\phi(\mathbf{x} \wedge \mathbf{y}) &= \int_{\mathbb{R}} \int_a^b \phi(x - y; \sigma_1^2)\, dy \int_c^d \phi(x - z; \sigma_2^2)\, dz\, dx \\
&= \int_c^d \int_a^b \phi(y - z; \tau^{-2})\, dy\, dz \\
&= \int_c^d \int_a^b \Phi'(\tau(y - z))\tau\, dy\, dz \\
&= \int_c^d \Phi(\tau(b - z)) - \Phi(\tau(a - z))\, dz \\
&= \frac{-1}{\tau}(m_\Phi(\tau(b - d)) - m_\Phi(\tau(a - d)) - m_\Phi(\tau(b - c)) + m_\Phi(\tau(a - c)))
\end{aligned}$$

and therefore, with $\sigma = \tau^{-1}$,

$$p_\phi(\mathbf{x} \wedge \mathbf{y}) = \sigma\left(m_\Phi(\tfrac{b-c}{\sigma}) + m_\Phi(\tfrac{a-d}{\sigma}) - m_\Phi(\tfrac{b-d}{\sigma}) - m_\Phi(\tfrac{a-c}{\sigma})\right)$$

as desired.

## B    MOVIELENS PSEUDOSPARSITY

The MovieLens dataset, while not truly sparse, has a large proportion of small probabilities which make it especially suitable for optimization by the smoothed model. The rough distribution of probabilities, in buckets of width $0.1$, is shown in Figure 1.

## C    MOVIELENS INITIALIZATION SENSITIVITY

We perform an additional set of experiments to determine the robustness of the smoothed box model to initialization. While the model is normally initialized randomly so that each box is a product of intervals that almost always overlaps with the other boxes, we would like to determine the models robustness to disjoint boxes in a principled way. While we can control initialization, we cannot always

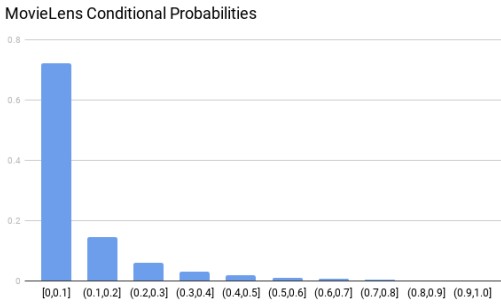

Figure 1: Distribution of probabilities in MovieLens Dataset.

control the intermediate results of optimization, which may drive boxes to be disjoint, a condition from which the original, hard-edged box model may have difficulty recovering. So, parametrizing the initial distribution of boxes with a minimum coordinate and a positive width, we adjust the width parameter so that approximately 0%, 20%, 50%, and 100% of boxes are disjoint at initialization before learning on the MovieLens dataset as usual. These results are presented in table 8. The smoothed model does not seem to suffer at all from disjoint initialization, while the performance of the original box model degrades significantly. From this we can speculate that part of the strength of the smoothed box model is its ability to smoothly optimize in the disjoint regime.

| Approx. % Disjoint | KL | | Pearson | | Spearman | |
|---|---|---|---|---|---|---|
| | Box | Smooth | Box | Smooth | Box | Smooth |
| 0% | 0.0147 | 0.0138 | 0.8775 | 0.8985 | 0.8768 | 0.8977 |
| 20% | 0.0172 | 0.0141 | 0.8668 | 0.8917 | 0.8608 | 0.8898 |
| 50% | 0.0182 | 0.0141 | 0.8613 | 0.8908 | 0.8551 | 0.8910 |
| 100% | 0.0346 | 0.0142 | 0.8401 | 0.8921 | 0.8167 | 0.8947 |

Table 8: Performance of the original box model and smoothed box model on MovieLens, as a function of different degrees of disjointness upon initialization.

## D    MODEL PARAMETERS

We give a brief overview of our methodology and hyperparameter selection methods for each experiment. Detailed hyperparameter settings and code to reproduce experiments can be found at `https://github.com/Lorraine333/smoothed_box_embedding`.

### D.1    WORDNET PARAMETERS

For the WordNet experiments, the model is evaluated every epoch on the development set for a large fixed number of epochs, and the best development model is used to score the test set. Baseline models are trained using the parameters of Vilnis et al. (2018), with the smoothed model using hyperparameters determined on the development set.

### D.2    IMBALANCED WORDNET PARAMETERS

We follow the same routine as the WordNet experiments section to select best parameters. For the 12 experiments we conducted in this section, negative examples are generated randomly based on the ratio for each batch of positive examples. We do a parameter sweep for all models then choose the best result for each model as our final result.

### D.3 FLICKR PARAMETERS

The experimental setup uses the same architecture as Vilnis et al. (2018) and Lai & Hockenmaier (2017), a single-layer LSTM that reads captions and produces a box embedding parameterized by *min* and *delta*. Embeddings are produced by feedforward networks on the output of the LSTM. The model is trained for a large fixed number of epochs, and tested on the development data at each epoch. The best development model is used to report test set score. Hyperparameters were determined on the development set.

### D.4 MOVIELENS PARAMETERS

For all MovieLens experiments, the model is evaluated every 50 steps on the development set, and optimization is stopped if the best development set score fails to improve after 200 steps. The best development model is used to score the test set.

