# OpenReview forum: "Smoothing the Geometry of Probabilistic Box Embeddings"
_ICLR.cc/2019/Conference_

### Official Review · AnonReviewer3 · 2018-10-17

**Rating:** 7
**Confidence:** 3

**Review:**

Post-rebuttal revision: All my concerns were adressed by the authors. This is a great paper and should be accepted.

------

The paper presents smoothing probabilistic box embeddings with softplus functions, which make the optimization landscape continuous, while also presenting the theoretical background of the proposed method well. The paper presents the overall idea beautifully and is very easy to follow. The overall idea of smoothed sotfplus boxes is well-founded, elegant and practical. The results on standard WordNet do not improve upon state-of-the-art, however imbalanced WordNet with abundance of negative examples gain remarkable improvements. Similarly in Flickr and MovieLens the method performs well. This paper presents a novel, theoretically well-justified idea with excellent results, and is likely going to be a high-impact paper.

An illustrating figure would still be nice to include, also for the convolutions of eq 2. The paper does not comment on running times, some kind of scalability comparison should be included since the paper claims that the model is easier to train.

The paper should clarify that the \prod in 3.3. meet and join definitions seems to refer to a set product, while the p(a) equation has a standard product (or does it?). What is the “a” in the p(a), should it be "p(x)” ?

I have trouble understanding eq 1: the difference inside the function is always negative, while the hinge function seems to clip negative values away. The definition of the m(x) is too clever, please clarify the function in more conventional notation.

---

> ### Author Response · Authors · 2018-11-27
> **Response**
>
> Thank you for your thoughtful review. Responses are included inline:
>
> > An illustrating figure would still be nice to include, also for the convolutions of eq 2.
>
> We agree that such a rendering will be helpful, and will add it to the paper.
>
> > The paper does not comment on running times, some kind of scalability comparison should be included since the paper claims that the model is easier to train.
>
> The ease of training leads to better results on certain data, rather than increased scalability --- both methods are applicable to large scale data, similar to other embedding methods. We added a new series of experiments testing robustness to different initialization regimes for the two models, which are included in the draft and detailed in our response to Reviewer #2.
>
> > The paper should clarify that the \prod in 3.3. meet and join definitions seems to refer to a set product, while the p(a) equation has a standard product (or does it?). What is the “a” in the p(a), should it be "p(x)” ?
>
> Your interpretation of the products is correct, and "a" was indeed a typo for "x." Thanks! We have fixed this in the draft and changed the definition to clarify the meaning of the products.
>
> > I have trouble understanding eq 1: the difference inside the function is always negative, while the hinge function seems to clip negative values away.
> > The definition of the m(x) is too clever, please clarify the function in more conventional notation.
>
> Thank you, there was a sign error. In the updated formula, the quantity inside the function can be positive or negative (negative if the hard boundaries of the boxes don't overlap at all). We've also switched the definition to use “min” and “max” rather than \wedge and \vee symbols, so it should be much clearer.

---

### Official Review · AnonReviewer2 · 2018-11-02

**Rating:** 8
**Confidence:** 4

**Review:**

This paper proposes a soft relaxation of the box lattice (BL) model of Vilnis et al. 2018 and applies it to several graph prediction tasks. Results are comparable to the BL model on existing artificially-balanced data but significantly better on more natural unbalanced data with a large number of negatives. The paper assumes some familiarity with the problem domain and existing works (there is not a lot of exposition for an unfamilar reader), but should be of strong interest to anyone working on embeddings or graph prediction.

The paper is well-written, with clear explanations of the desired properties of the model and a concise set of experiments that are easy to follow. The strongest result is that on unbalanced WordNet, while the Flickr and MovieLens results are a little less clear but do show that this technique does not cause any loss in performance.

A few points of feedback:

- Missing citation / comparison: https://arxiv.org/pdf/1804.01882.pdf (Ganea et al. 2018) is an alternative way of generalizing order embeddings. They also report very high numbers on WordNet, though I'm not sure they are directly comparable.

- The Gaussian relaxation (Eq. (2) and (3)) defines a particular length scale, \sigma. It's not clear if this is also implicit in the softplus derivation (by analogy with Eq. (4), should we assume that it approximates the \sigma = 1 case?). What effect does this have on the embedding space? Without it, it would seem that the normal BL model is scale invariant, which might be a desirable property for representing hierarchical data.

- The main thrust of section 5.2 is that smoothed box embeddings retain better performance with increasing numbers of negatives. Could you include the ratio of positive / negative examples on the Flickr dataset, and some measure of the distribution of P(A|B) values on MovieLens to get a sense of how these datasets compare?

- Flickr data: what is the encoder model that produces the embeddings here, and how does it handle unseen captions? (Why would we expect the smoothed box model to handle unseen captions better?)

- There's a strong emphasis on how smoothing makes training easier. Do you have any metrics to directly support this, such as variance under random restarts?

- In the abstract and introduction, it's easy to gloss over "inspired by" and assume that the actual model is a Gaussian convolution. Could be more direct here that it's a softplus approximation.

---

> ### Author Response · Authors · 2018-11-27
> **Response**
>
> Thank you for the review. We will reply in detail to each point inline:
>
> > Missing citation / comparison: https://arxiv.org/pdf/1804.01882.pdf (Ganea et al. 2018) is an alternative way of generalizing order embeddings.
> > They also report very high numbers on WordNet, though I'm not sure they are directly comparable.
>
> This is indeed a very related paper. Our work differs from hyperbolic embeddings in a couple of ways. First, by virtue of being a probabilistic model, the box model can score complex multivariate queries including negated variables. Secondly, the box structure is more suitable for general DAG embedding, as opposed to a hyperbolic model where the constant negative curvature strongly biases the model towards trees. The numbers are not directly comparable, but we will add this to related work, thank you.
>
> > The Gaussian relaxation (Eq. (2) and (3)) defines a particular length scale, \sigma.
> > It's not clear if this is also implicit in the softplus derivation (by analogy with Eq. (4), should we assume that it approximates the \sigma = 1 case?).
> > What effect does this have on the embedding space? Without it, it would seem that the normal BL model is scale invariant, which might be a desirable property for representing hierarchical data.
>
> The \sigma parameter is absorbed into the constant \rho in the softplus approximation to the Gaussian (Proposition 1), which differs from \sigma by the factor 1/1.702 given there.  In practice, this is tuned as a global temperature for the softplus, but it is not particularly important when normalizing the space by the global coordinatewise minimum and maximum, as explained at the end of section 4.2 (this detail is probably the most important practical answer to your question). The scale invariance question is interesting. In order to solve the problem of sparse gradients, our solution sacrifices scale invariance. While scale invariance is desirable in theory, it has been known to cause instability in other contexts, such as perceptron vs. hinge loss learning, and perhaps the “scale” of the “soft edges” could be viewed as a type of margin, as well as solving the problem of sparse gradients.
>
> > The main thrust of section 5.2 is that smoothed box embeddings retain better performance with increasing numbers of negatives.
> > Could you include the ratio of positive / negative examples on the Flickr dataset, and some measure of the distribution of P(A|B) values on MovieLens to get a sense of how these datasets compare?
>
> Since the Flickr dataset consists of denotational entailment probabilities between (possibly unseen) pairs of sentences, none of the train or test probabilities are exactly 0 (negative examples). However, many such pairs have a conditional probability below 0.1, with a ratio of about 13:1. Movielens is similarly pseudosparse, not truly sparse, with a similarly large majority of its probabilities taking values below 0.1. We have added a histogram showing the distribution of these probabilities in the appendix.
>
> > Flickr data: what is the encoder model that produces the embeddings here, and how does it handle unseen captions? (Why would we expect the smoothed box model to handle unseen captions better?)
>
> The encoder model is a single-layer LSTM with the same specifications as used in Lai and Hockenmaier 2017 and Vilnis et al. 2018. It handles unseen captions by composing token embeddings with the RNN. We have updated the draft to make this clear. As for why the soft box model improves on unseen captions more than the other tasks, it may simply be a question of there being more room to improve (the previous SOTA held-out KL divergence is about twice as large for unseen captions than for the other categories, for example.) It would be interesting to explore this further.
>
> (continued in next comment)

---

> > ### Author Response · Authors · 2018-11-27
> > **Response Continued**
> >
> >
> > > There's a strong emphasis on how smoothing makes training easier. Do you have any metrics to directly support this, such as variance under random restarts?
> >
> > We do not see much variance in terms of outcome when changing only the random seed. In terms of ease/robustness of training, our experiments on imbalanced wordnet give evidence that the soft box model is more robust in the regime of sparse *training* data. However, we have updated the draft with a new series of experiments on MovieLens. In the appendix, we’ve added experiments that demonstrate the greatly decreased sensitivity of the soft box model when picking distributions for box initialization such that the boxes start off with roughly 0%, 20%, 50%, and 100% of boxes disjoint --- regimes in which the hard box model experiences much greater degradation in performance. Although we can control our initialization, we can't necessarily control the intermediate stages of learning, during which boxes may become disjoint, so this may give some useful insight.
> >
> > NOTE: When performing this comparison, we found a difference between the criteria to establish development set convergence in the POE, box, and soft box experiments on MovieLens and the criteria used by the (complex) bilinear baseline models. These criteria (number of steps without development set improvement) are given in the appendix. This led us to update the results (in Table 5) for POE, box, and soft box, with the best performing model (our proposed soft box model) improving by an absolute point of Spearman and Pearson's rho compared to the old tuning regime. Additionally, the hard box model outperforms all other models besides the soft box model. The soft box model outperforms it in KL and Pearson by a similar absolute margin as before, but its previous advantage of ~2.9 points of Spearman's rho over the hard box model is now only ~2.1 points. This difference in development set stopping criteria was not present in any other experiments.
> >
> >
> > > In the abstract and introduction, it's easy to gloss over "inspired by" and assume that the actual model is a Gaussian convolution. Could be more direct here that it's a softplus approximation.
> >
> > The model is also modified to take pointwise min and max inside the softplus, in order to maintain idempotency, as described in the second half of section 5.2. We updated this section with a clearer description. Since we not only approximate the Gaussian with a logistic, but also modify the equation to preserve the necessary idempotency (by analogy to the zero-temperature limit), "softplus approximation" might not be sufficient to describe the entire model. We should still try to make this part of the abstract clearer in some way.

---

### Official Review · AnonReviewer1 · 2018-11-03
**Nice idea and good improvement on benchmarks**

**Rating:** 8
**Confidence:** 3

**Review:**

The paper proposes a method for learning embedding of hierarchies. Specifically, the paper builds on a a geometrically inspired embedding method using box representations. The key contribution of the paper is facilitating optimization of these models by gradient based methods, which eventually leads to improved accuracy on relevant benchmark data (on par or beyond SOTA). The observation is that when two boxes are disjoint in the model but have overlap in the ground truth, no gradient can flow to the model to correct the problem (which is happens in case of sparse-data.

To alleviate the above problem, the paper proposes smoothing the model. That is, transforming the original model constructed from indicator functions (hence difficult to optimize by gradient based method) to a smooth differentiable function by diffusing the landscape. The diffusion process corresponds to convincing the objective function with the Gaussian kernel.

I find the idea of converting such combinatorial problems to differentiable, specially when gradient methods can succeed in optimizing them afterward, very fascinating. I believe this paper is taking a theoretically sound path to construct the differentiable form of the originally non-differentiable problem. As the authors find, the smoothed function leads to improved performance against SOTA on relevant benchmark data such as WordNet hypernymy, Flick caption entailment and MovieLnes market basket data.

One downside of the current submission is that the details of optimization are now provided at all. What algorithm do you use to optimize the objective function? What are the hyper parameters? What value of sigma (for diffusion) do you use [or maybe you use the continuation method to gradually anneal sigma from large toward zero?). These are important details that I ask the authors to include.

Also, I think some graphical illustration of the embedding would be very helpful, perhaps something like Figure 2 of "Probabilistic Embedding of Knowledge Graphs with Box Lattice Measures". I hope such illustration is added to the submission.

---

> ### Author Response · Authors · 2018-11-27
> **Response**
>
> Thank you for the thoughtful review.
>
>  - We use Adam to perform the optimization, using the default settings given in the Adam paper for momentum / decay / ridge terms, with learning rates given in the appendix of the submission. We have also updated the appendix with more hyperparameter details, and plan to release code before publication.
>
>  - The temperature / bandwidth hyperparameter is always set equal to 1.0. We address this also in our response to Reviewer #2 --- since in all experiments aside from Flickr, we divide each dimension by the global maximum across boxes, this seems to avoid scale issues.
>
>  - We agree that a figure illustrating the geometric intuition would be helpful, and will add a rendering in a future draft.

---

### Public Comment · (anonymous) · 2018-11-01
**Missing Reference**

This paper seems like a great idea. However, I believe the paper misses an important reference on the WordNet task. According to Hierarchical Density Order Embeddings (Athiwaratkun, 2018) https://arxiv.org/pdf/1804.09843.pdf, their score for hypernym prediction on WordNet test split is 92.3 which is a bit higher than the paper's reported scores.

---

> ### Author Response · Authors · 2018-11-02
> **Reply to Missing Reference**
>
> Hi, thanks for the comment! We actually do cite the Hierarchical Density Order Embedding paper from Athiwarakun et al. in the introduction section. The original box lattice paper from Vilnis et al. also reports the density model result of 92.3 accuracy in their wordnet table. Box embeddings get a very similar score on this task, so we only include that result, since the aim of the experiment is to compare the softbox and hard box models and not to demonstrate a new state of the art. There are also some questions about whether the density embeddings use exactly the same dataset split, or just the same method of generating negative examples, which we have not been able to determine. Hope this helps!

---

### Meta-Review · Area_Chair1 · 2018-12-13

**Confidence:** 4
**Recommendation:** Accept (Oral)

**Metareview:**

The manuscript presents a promising new algorithm for learning geometrically-inspired embeddings for learning hierarchies, partial orders, and lattice structures. The manuscript builds on the build on the box lattice model, extending prior work by relaxing the box embeddings via Gaussian convolutions. This is shown to be particularly effective for non-overlapping boxes, where the previous method fail.

The primary weakness identified by reviewers was the writing, which was thought to be lacking some context, and may be difficult to approach for the non-domain expert. This can be improved by including an additional general introduction. Otherwise, the manuscript was well written.

Overall, reviewers and AC agree that the general problem statement is timely and interesting, and well executed. In our opinion, this paper is a clear accept.